# A Current and Newly Proposed Artificial Intelligence Algorithm for Reading Small Bowel Capsule Endoscopy

**DOI:** 10.3390/diagnostics11071183

**Published:** 2021-06-29

**Authors:** Dong Jun Oh, Youngbae Hwang, Yun Jeong Lim

**Affiliations:** 1Department of Internal Medicine, Dongguk University Ilsan Hospital, Dongguk University College of Medicine, Goyang 10326, Korea; mileo31@naver.com; 2Department of Electronics Engineering, Chungbuk National University, Cheongju 28644, Korea; ybhwang@chungbuk.ac.kr

**Keywords:** artificial intelligence, automatic detection, capsule endoscopy, reading software

## Abstract

Small bowel capsule endoscopy (SBCE) is one of the most useful methods for diagnosing small bowel mucosal lesions. However, it takes a long time to interpret the capsule images. To solve this problem, artificial intelligence (AI) algorithms for SBCE readings are being actively studied. In this article, we analyzed several studies that applied AI algorithms to SBCE readings, such as automatic lesion detection, automatic classification of bowel cleanliness, and automatic compartmentalization of small bowels. In addition to automatic lesion detection using AI algorithms, a new direction of AI algorithms related to shorter reading times and improved lesion detection accuracy should be considered. Therefore, it is necessary to develop an integrated AI algorithm composed of algorithms with various functions in order to be used in clinical practice.

## 1. Introduction

Small bowel capsule endoscopy (SBCE) was first performed in humans in 2000 [1]. Since then, it has become the first modality in the diagnosis of various small bowel diseases such as obscure gastrointestinal (GI) bleeding, Crohn’s disease, small bowel tumor or polyposis syndrome, and celiac disease [2,3,4]. With the advancement of technology, the function of capsule endoscopy, such as battery life and optical performance, have improved [5,6]. However, the long reading time (often more than 1 h) of SBCE has yet to be solved. The length of the small bowel is known to be about 6–8 m [7]. The battery time is about 8 to 12 h and about 50,000 to 100,000 images are captured during an SBCE examination [2,8]. The European Society of Gastrointestinal Endoscopy (ESGE) guidelines recommended a reading rate at a maximum of 10 frames per second in a single view mode [9]. However, reading capsule images for a long time is inevitably boring and burdensome for the clinician. There is also a risk of error resulting from eyestrain.

Therefore, several options have been proposed to reduce the burden of SBCE reading for the clinician. The ESGE guidelines suggest the use of skilled nurses or technicians for pre-reporting [9]. However, this needs additional manpower. Software tools have also been developed to aid in SBCE reading. There have been studies on the suspected bleeding indicator (SBI), which detected obscure GI bleeding, i.e., the most common indication of SBCE [10]. In addition, one study reported that the “QuickView mode”, which shows the selected image in the viewer software, reduced the reading time compared to the conventional reading by a clinician (e.g., in short, conventional reading) [11]. However, these reading tools have a limitation in that they are less accurate than conventional reading [11,12]. Currently, lesion detection using an AI algorithm has emerged as a way to save reading time while maintaining the accuracy of lesion detection compared to conventional reading [13].

Therefore, we summarized the current AI algorithms for SBCE reading and suggested new trends to give insight to researchers and clinicians studying this field.

## 2. Application of Artificial Intelligence (AI) into the Reading of Small Bowel Capsule Endoscopy (SBCE)

Since 2010, deep learning and convolutional neural network (CNN) algorithms have been used in medicine. Currently, the CNN algorithm is being used as the dominant AI algorithm in the field of medical image reading [13]. As previously mentioned, it takes a lot of time and effort to read a capsule endoscopy. However, among tens of thousands of SBCE images, lesions appear only in a few [14]. To solve this problem, it is essential that reading software can accurately detect small bowel lesions. Therefore, the need for lesion detection using an AI algorithm has increased in SBCE reading.

### 2.1. Automatic Detection of Small Bowel Lesions

In 2016, a study was published that focused on bleeding detection using an AI (especially CNN) algorithm in SBCE images [15]. Since then, many studies have been published that use an AI algorithm to read SBCE images to detect of various small bowel lesions, including inflammatory lesions (such as erosions and ulcers) [16,17,18,19], vascular lesions (such as bleeding and angioectasia) [20,21,22], and protruding lesions [23]. In these studies, high sensitivity and specificity for lesion detection were confirmed and the feasibility of an AI algorithm for SBCE reading was demonstrated.

However, in a clinical SBCE video, various lesions were shown to exist at the same time or in several places throughout the video. Therefore, studies are needed to develop and verify AI algorithms that can locate various small bowel lesions for use in clinical practice. In one large scale study [24], 158,235 images that contained two normal variants and eight abnormal lesions were used to train an AI algorithm. Then, the validation of an AI algorithm was performed with images not included in the training set. Compared to the conventional reading, the AI-assisted reading showed higher sensitivity and specificity for lesion detection (sensitivity of 99.9% and specificity of 99.9% in AI-assisted reading vs. sensitivity of 74.6% and specificity of 76.9% in conventional reading). The reading time was also reduced by about 94% in the AI-assisted reading compared to that in the conventional reading (96.6 min in conventional reading to 5.9 min in AI-assisted reading). In a multicenter study [25], 66,028 images containing normal mucosa and various lesions were used to train an AI algorithm. The overall accuracy of lesion detection was 98% when reading was performed using an AI algorithm. This showed a higher accuracy compared to that of 89% in “QuickView mode”, which was used as a control. In particular, for protruding lesions, the detection accuracy of an AI algorithm significantly improved over the “QuickView mode” (99% vs. 80%). In another study [26], 7556 images containing hemorrhagic lesions and ulcerative lesions (i.e., the most common lesions in SBCE images) were used to train an AI algorithm. A lesion detection accuracy of 96.83% and a sensitivity of 97.6% were confirmed when SBCE reading was performed using an AI algorithm. In another AI-assisted reading study [27], 60,000 images of significant and insignificant lesions were divided by binary classification and used to train an AI algorithm. In total, 20 SBCE cases were externally validated by experts and trainees using conventional reading and AI-assisted reading, respectively. In the external validation test for all 20 SBCE cases, the overall lesion detection rate increased from 29.5% with conventional reading to 63.1% with AI-assisted reading. Moreover, when AI-assisted reading was applied to trainees, the total reading time for 20 SBCE cases reduced by 64% compared to the conventional reading (1621 min with the conventional reading vs. 587 min with AI-assisted reading). In a study that used 39,963 images containing normal and various lesions for training an AI algorithm, area under the curve (AUC) values for detecting inflammatory lesions, vascular lesions, and tumorous lesions were all 0.95 or higher [28] (Table 1).

To date, several studies using an AI algorithm for SBCE reading focused on detecting lesions. For automatic lesion detection in selected and single still images, an AI algorithm showed high accuracies. However, studies on AI-assisted reading for images obtained from a of full-length capsule endoscopy are still lacking. In one study [29], 20 full-length SBCE videos, including erosions and ulcers, were read using an AI algorithm. The AI-assisted reading shortened the reading time while maintaining the detection rate compared to the conventional reading. However, this study had a limitation that an AI algorithm did not: it read multiple lesions, such as vascular lesions and protruding lesions. A recent multicenter study [25] was conducted on the detection of various lesions using an AI algorithm at the full-length SBCE video level. This study showed a high detection rate in per-patient analysis, but per-lesion analysis could not be carried out. In addition, it is absolutely necessary to confirm the actual performance of an AI algorithm through a prospective study.

### 2.2. Automatic Classification of Small Bowel Cleanliness

Although the stomach and colon can be cleaned by suction and washing in a wire endoscopy, bowel cleansing cannot be actively performed in a SBCE. Proper small bowel preparation affects the quality control and lesion detection of SBCE. Therefore, adequate bowel cleanliness is important during the SBCE examination [30,31]. Although small bowel cleanliness scales have been developed [32,33], they also cannot objectively represent the whole small bowel cleanliness. It has also been shown that the intra-observer reproducibility was low when classifying small bowel cleanliness [34]. To increase the intra-observer reproducibility and assess the bowel cleanliness as an objective indicator, several studies have been conducted to evaluate small bowel cleanliness using an AI algorithm. 

In one study [35], 55,293 images were classified into dirty and clean images according to a 4-level scale and used to train an AI algorithm. In total, 30 SBCE cases were tested with an AI algorithm and the accuracy of the small bowel cleanliness assessment was confirmed to be 95.2%. In another study [36], 600 normal small bowel images were classified into adequate and inadequate cleanliness according to a 10-point scale and used to train with an AI algorithm. Adequacy evaluation of small bowel cleanliness showed a sensitivity of 90.3%, a specificity of 83.3%, and an accuracy of 89.7% when an AI algorithm was tested using 156 SBCE cases. In a recent study [37], an AI algorithm was trained using 71,191 images that classified bowel cleanliness according to a five-step scoring method. Then, an automated scoring of small bowel cleanliness was conducted by using a trained AI algorithm. The average cleanliness score was 4.0 for the adequate group and 2.9 for the inadequate group (*p* < 0.001). When the cut-off value of cleanliness score was 3.25, the AUC of small bowel cleanliness was found to be 0.977.

### 2.3. Automatic Compartmentalization of Small Bowel

The main indication of SBCE obscures GI bleeding [2,3]. Therefore, in most cases, upper endoscopy and colonoscopy are performed before SBCE [38]. In other words, the area from oral cavity to second portion of duodenum was already confirmed by wire endoscopy. However, when reading the SBCE in clinical practice, one must first examine the images where the capsule stays in the stomach. During a SBCE, the mean gastric transit time is about 50 min [39]. However, in about 6% of SBCE cases, the capsule stays in the stomach for more than 90 min (delayed gastric transit) or fails the duodenal transit [40]. Therefore, even if the time when capsule passes through the pylorus is accurately identified, the clinician can reduce some of the reading time. In a recent study using OMOM capsule endoscopy device (Jinshan, Chongqing, China) [41], the first duodenal images were used to train an AI algorithm to identify the duodenal transition of the capsule. AUC of 0.984, sensitivity of 97.8%, and specificity of 96.0% were confirmed in the duodenum transit of the capsule. The difference between the actual transit time and the AI determined transit time was mostly within 8 min. The completion of the SBCE study is related to the quality of the SBCE [30]. Therefore, additional research is needed to confirm the cecal transition of the capsule using an AI algorithm.

## 3. New Proposals for Using an AI Algorithm in Clinical Practice

AI-assisted reading of SBCE is essential for clinicians [42]. However, in order for the AI algorithm to be used commercially, complementary directions are still needed [43]. Therefore, not only reinforcing automatic lesion detection but developing AI algorithms from novel perspectives should be considered.

### 3.1. Automatic Filtering of Normal Images

It can be difficult to determine whether or not a lesion is significant. However, the main reason that clinicians feel burdened in SBCE reading is the long reading time. When SBCE reading, most images are normal mucosa or normal variants such as bile, bubble, and debris. Lesions usually appear only in a few frames. Therefore, if an AI algorithm can accurately recognize and exclude the definite images of normal mucosa and normal variants, the clinician can focus on lesion detection and thus reduce the reading time.

With a similar concept, various software tools have been studied to reduce the reading time in SBCE reading. The “QuickView mode”, available in the RAPID Software of PillCam (Medtronic, Minneapolis, USA), can show selected images. In one study [44], “QuickView mode” reduced the SBCE reading time by about 70% compared to conventional reading (59.8 min in conventional reading vs. 16.3 min in “QuickView mode” reading). However, the disadvantage is that the diagnostic miss rate was higher than the conventional reading (12% in “QuickView mode” reading vs. 1% in conventional reading). EndoCapsule (Olympus, Tokyo, Japan) developed “Omni mode”, a rapid reading tool that can reduce the reading time by removing duplicate images [45]. In a clinical study [46], “Omni mode” reduced reading time by about 40% compared to conventional reading (mean 42.5 minutes in conventional reading vs. mean 24.6 minutes in “Omni mode” reading). Besides, a reading accuracy of “Omni mode” was not significantly different from a conventional reading.

By incorporating an AI algorithm into the concept of these reading tools, the diagnostic miss rate can be reduced based on the high accuracy of AI. Like P classification [47], normal mucosa and abnormal lesion images can be used to train an AI algorithm for summarizing and filtering of normal or duplicated images. Then, a trained AI algorithm can differentiate between normal and abnormal images according to the probability score of a lesion. Moreover, in the case of a blurry or poorly cleaned SBCE image, it is often impossible to determine whether there is a lesion within the image. If an AI algorithm for filtering normal, duplicate, and blurry images is developed, the clinician will be able to focus more on images with suspected lesions.

### 3.2. Automatic Reconfirmation of Small Bowel Lesion

Until now, most studies that used an AI algorithm to detect multiple lesions focused on significant lesions for training and testing. However, normal variants such as air bubbles, concentrated bile, debris, and light reflections can be confused with lesions such as ulcers, bleeding, and subepithelial lesions. In SBCE reading that uses an AI algorithm, many false positives may be generated due to these normal variants (Figure 1). In an actual full-length SBCE video, it can be difficult to determine if it is a significant lesion or not based on a single image. In this case, the clinician identifies the front and back frames of the suspected lesion image to confirm whether it is a definite lesion or not. Therefore, an AI algorithm reading at a multi-image level rather than a single image level can be helpful as a method to reduce detection errors. If a lesion is suspected in a single image during an AI algorithm reading, measuring the probability of the lesion again at the multi-image level will reduce false positives compared to single image detection.

### 3.3. New Consensus on the Small Bowel Lesions

Capsule endoscopy structured terminology (CEST) was proposed in 2005 as a standardized terminology in SBCE [48]. However, it is not easy to use it in clinical practice due to its complexity. Moreover, inter-observer variability related to the lesion can occur for each clinician, except for clear lesions such as deep ulcers and active bleeding [49]. An ambiguous detection such as a simple hyperemic spot or lymphofollicular hyperplasia may not be related to the small bowel disease. To use the AI algorithm in medical aspects, consensus on the clinically significant small bowel lesions is required.

## 4. Conclusions

SBCE reading via an AI algorithm showed high accuracy for lesion detection. However, it essential to study fully reliable AI algorithms that read full-length capsule endoscopy video. In addition, multicenter prospective studies should be conducted. Moreover, to read SBCE videos at the same level as a clinician, integrated AI algorithm that include automatic lesion detection, small bowel compartmentalization, normal image filtering, and lesion confirmation via multi-image level is required (Figure 2). Therefore, if we can solve the limitations, SBCE reading using an AI algorithm can be realized in actual clinical practice.

## Figures and Tables

**Figure 1 diagnostics-11-01183-f001:**
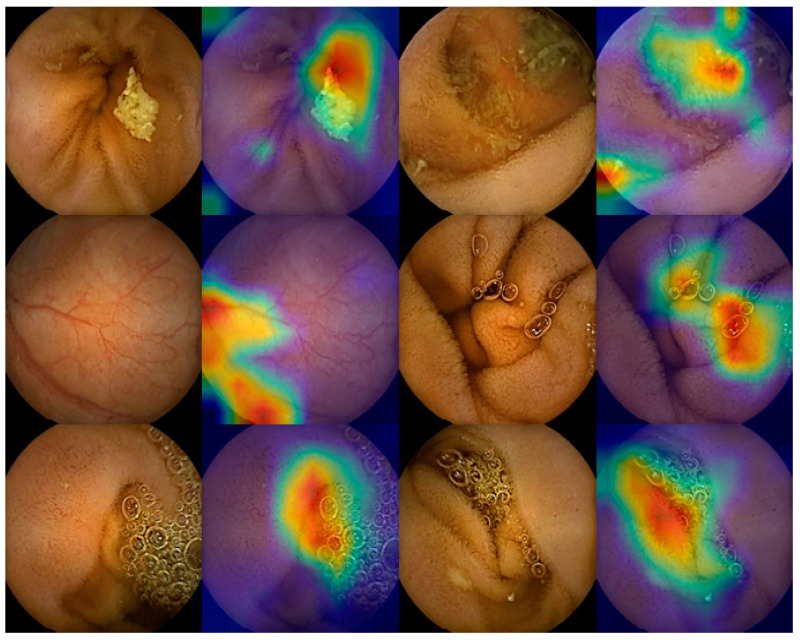
Examples of detection error in the binary classification of artificial intelligence (AI) reading. The bile, vascular structure, bubble, and normal mucosal fold were mistaken for significant lesions in AI reading.

**Figure 2 diagnostics-11-01183-f002:**
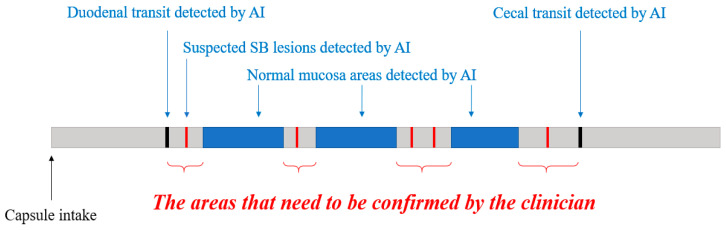
Proposed schematic diagram of an integrated artificial intelligence (AI) algorithm used to read a small bowel capsule endoscopy. First, an AI algorithm identifies the duodenal image and cecal image in the full-length capsule video. Then, the AI algorithm differentiates the suspected lesions from the normal mucosa in the entire small bowel. Finally, the clinician confirms the significant lesion. An integrated AI algorithm composed of AI algorithms in charge of each function will be more useful in actual clinical practice.

**Table 1 diagnostics-11-01183-t001:** Summary of automatic detection of various lesions by using an AI algorithm for reading of small bowel capsule endoscopy.

Author (CNN System)	Lesion Categories (Trained Images) Validation and/or Test (Images)	Results
Ding et al. [24] (ResNet)	2 normal variants lymphangiectasia, lymphatic follicular hyperplasia 8 abnormal lesions inflammation, ulcer, bleeding, polyp vascular disease, protruding lesion, diverticulum, parasite (Total 158,235 trained images)	1. Overall sensitivity 99%. 2. Overall specificity 100%. 3. Shorter reading times than conventional reading. (*p* < 0.001)
	5000 cases (113,268,334 images)	
Aoki et al. [25] (SSD + ResNET50)	Mucosal breaks (5360 images) Angioectasia (2237 images) Protruding lesions (30,584 images) Blood content (6503 images)	1. Detection rate was 100%, 97%, 99% and 100% for each lesion.
	379 cases (5,050,226 images)	
Otani et al. [28] (RetinaNet)	Erosions and ulcers (398 images) Vascular lesions (538 images) Tumors (4590 images)	1. AUC 0.996 at inflamed 2. AUC 0.950 at vascular 3. AUC 0.950 at tumors
	29 cases (14,867 images) in external validation	
Park et al. [27] (Inception-Resnet-V2)	Inflamed mucosa Atypical vascularity, or bleeding (Total 60,000 images)	1. Overall AUC 0.998. 2. Shorter the reading time for trainees (*p* = 0.029)
	20 cases (210,100 images) in external validation	
Hwang et al. [26] (VGGNet and Grad-CAM)	Hemorrhagic lesions Ulcerative lesions (Total 3778 images ^1^)	1. Overall AUC 0.9957 2. Sensitivity 96.95% 3. Specificity 97.13%
	162 cases (5760 images)	

^1^ 30,224 augmented (×8) image was used for training dataset.

## Data Availability

Not applicable.

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
