# Peer review of "A Current and Newly Proposed Artificial Intelligence Algorithm for Reading Small Bowel Capsule Endoscopy"

_diagnostics, 2021, doi:10.3390/diagnostics11071183_

Round 1
Reviewer 1 Report
Thank you very much for the opportunity to read this manuscript. Small bowel capsule endoscopy (SBCE) is the only non-invasive method of assessing the small intestine mucosa available. Enhancing it through artificial intelligence algorithms (AI) is extremely necessary to reduce study interpretation time and the risk of diagnostic errors. As in classic endoscopy, AI support is the future of diagnostics. This paper is a summary of research on the usefulness of AI in SBCE. The manuscript is well written in English. Its limitation is that it does not add anything new. We all know that more research and better AI are needed. However, it should be noted that there is not much more to say on this issue because there is not enough data. Emphasizing this is the paper’s main goal and deserves publication as another voice in this discussion and encouragement for further research.
Below are some minor comments to the Authors:
- The final sentence of the Abstract needs to be corrected to make it more transparent.
- I suggest an alphabetical order of keywords (line 20).
- Some SBCEs have a battery life of more than 10 hours (line 30). In my practice with Olympus Endocapsule, it is 16 hours, but please check the manufacturer’s documentation - nominally, it is probably 12 hours.
- Not only boring, please highlight the risk of error resulting from eyestrain (line 33).
Author Response
Thank you very much for giving the opportunity of revision. The authors appreciate the critical and thoughtful comments. We did our best to write the reply for your comments. And we used the "track changes" function to revise the manuscript and make it easy for editors and reviewers to check.
Point 1: The final sentence of the Abstract needs to be corrected to make it more transparent.
Response 1: We changed from “If an integrated AI algorithm consisting of several SBCE reading algorithms is developed, the AI algorithm for reading of SBCE will be able to be used in clinical practice sooner.” to “Therefore, it is necessary to develop an integrated AI algorithm composed of algorithms with various functions in order to be used in clinical practice.”
Point 2: I suggest an alphabetical order of keywords (line 20).
Response 2: We changed keywords according to alphabetical order. (Artificial intelligence; Automatic detection; Capsule endoscopy; Reading software)
Point 3: Some SBCEs have a battery life of more than 10 hours (line 30). In my practice with Olympus Endocapsule, it is 16 hours, but please check the manufacturer’s documentation - nominally, it is probably 12 hours.
Response 3: We confirmed that Olympus ENDOCAPSULE 10 system has a minimum of 12 hours battery life. We changed from “The battery time is about 10 hours” to “The battery time is about 8 to 12 hours”
Point 4: Not only boring, please highlight the risk of error resulting from eyestrain (line 33).
Response 4: We agree with your opinion. We added the sentence “There is also the risk of error resulting from eyestrain” in line 33

Reviewer 2 Report
English should be polished (for instance: page 1, line 10, page 7, line 223).
There are several unclear sentences (page 2, lines 52-53, page 2, lines 89-91, page 4, lines 143-145, page 4, lines 147-150), which should be re-written.
References numbering should be corrected (45-49 are doubled).
Author Response
Thank you very much for giving the opportunity of revision. The authors appreciate the critical and thoughtful comments. We did our best to write the reply for your comments. And we used the "track changes" function to revise the manuscript and make it easy for editors and reviewers to check.
Point 1: English should be polished (for instance: page 1, line 10, page 7, line 223).
Response 1:
1) Page 1, Line 10: We changed from “Small bowel capsule endoscopy (SBCE) is the most useful methods for diagnosing small bowel mucosal lesions” to “Small bowel capsule endoscopy (SBCE) is one of the most useful methods for diagnosis of small bowel mucosal lesions.”
2) Page 1, Line 37: We changed from “Software tools that can help with reading the SBCE have also been developed.” to “Software tools have also been developed to aid in SBCE reading.”
3) Page 6, Line 212: We changed from “Currently, the accuracy of lesion detection is high in SBCE reading using an AI algorithm” to “Currently, SBCE reading using an AI algorithm showed high accuracy in lesion detection.”
4) Page 7, Line 222~223: We changed from “First, the AI algorithm identifies the transition of duodenum and cecum in the full-length capsule video, and the small bowel arear is specified.” to “First, an AI algorithm identifies first duodenal image and first cecal image in the full-length capsule video.”
5) We corrected wrong grammar and words throughout manuscript.
Point 2: There are several unclear sentences (page 2, lines 52-53, page 2, lines 89-91, page 4, lines 143-145, page 4, lines 147-150), which should be re-written.
Response 2:
1) Page 2, Line 52-53: We changed from “Among tens of thousands of SBCE images, the lesions usually appear a small portion [14]. Reading software that can accurately detect lesions in the SBCE images is essential” to “As previously mentioned, it takes a lot of time and effort to read a capsule endoscopy. However, among tens of thousands of SBCE images, the lesions usually appear in a few [14]. In order to solve this problem, reading software that can accurately detect small bowel lesions is essential.”
2) Page 2, Line 89-91: We changed from “Also, in the cases of AI-assisted reading which performed in trainees, the total reading time was reduced by 64% compared to the conventional reading (1,621 mins with the conventional reading vs. 587 mins with AI-assisted reading for 20 SBCE cases).” to “Also, when AI-assisted reading was applied to trainees, the total reading time for 20 SBCE cases was reduced by 64% compared to the conventional reading (1,621 mins with the conventional reading vs. 587 mins with AI-assisted reading).”
3) Page 4, Line 143-145: We changed from “In a recent study [41], a trained AI algorithm was identified the duodenal transit of capsule using OMOM device (Jinshan, Chongqing, China).” to “In a recent study using OMOM capsule endoscopy device (Jinshan, Chongqing, China) [41], the first duodenal images were used to train an AI algorithm to identify duodenal transition of capsule.”
4) Page 4, Line 147-150: We changed from “If the cecal transition of the capsule can be confirmed through additional researches, it will be possible to quickly confirm whether a complete or incomplete study for relation to the quality of SBCE [30].” to “Completion of the SBCE study is related to the quality of the SBCE [30]. Therefore, additional researches are also needed to confirm the cecal transition of capsule using an AI algorithm.”
Point 3: References numbering should be corrected (45-49 are doubled).
Response 3: We corrected references numbering.
